# LEARNING FLAT LATENT MANIFOLDS WITH VAES

## ABSTRACT

Latent-variable models represent observed data by mapping a prior distribution over some latent space to an observed space. Often, the prior distribution is specified by the user to be very simple, effectively shifting the burden of a learning algorithm to the estimation of a highly non-linear likelihood function. This poses a problem for the calculation of a popular distance function—the geodesic between data points in the latent space—as this is often solved iteratively via numerical methods. These are less effective if the problem at hand is not well captured by first or second-order approximations. In this work, we propose less complex likelihood functions by allowing complex distributions and explicitly penalising the curvature of the decoder. This results in geodesics which are approximated well by the Euclidean distance in latent space, decreasing the runtime by a factor of 1,000 with little loss in accuracy. Additionally, we apply our method to a state-of-the-art tracking algorithm using real world image data, showing that our unsupervised method performs similar to supervised learning methods.

## 1 INTRODUCTION

Latent-variable models (LVMs) are a viable tool in data analysis: a set of observations is explained by a simpler set of latent variables in conjunction with a map from the latent space to the space of observation. Methods from this family (e.g., principal component analysis (Wold et al., 1987), non-negative matrix factorisation (Lee & Seung, 2001), generalised discriminant analysis (Baudat & Anouar, 2000), etc.) are standard tools, serving either as feature extractors for subsequent data processing pipelines, density estimators or dimensionality reducers for visualisation.

Despite the maturity of the field, research has far from halted. While kernel methods (KernelPCA (Schölkopf et al., 1997), KernelNMF (Li & Ding, 2006), etc.) have been used to improve the applicability of LVMs to data inhibiting non-linear phenomena, neural formulations such as the variational autoencoder (VAE) (Kingma & Welling, 2014; Rezende et al., 2014) or the generative adversarial network (GAN) (Goodfellow et al., 2014) have become popular recently, especially due to their enormous success on modelling natural images. Here, a simple prior distribution (such as a multivariate standard normal or a uniform distribution) is mapped to the space of observations by means of a powerful deep neural network. A learning algorithm then finds weights for that neural network such that the data distribution is approximated well. In case of GANs, this is a minimax game, while the evidence lower bound is maximised in the case of VAEs.

If the data distribution is relatively complicated and the prior is relatively simple, the map from the latent to the observable space, the decoder, has to be sufficiently complex. In fact, it has to mimic the inverse CDF in parts. This results in highly non-linear neural networks. As an example, the separation of two modes in the probability landscape has to be implemented by a flat CDF, which in turn requires an infinitely steep inverse CDF. Not only does this often pose difficulties for gradient-based learning. It also hinders the calculation of geodesics, the shortest paths from one point to another, as measured by the rate of change in the decoder along the path in latent space. Geodesics are often solved numerically, i.e. through a gradient-based optimisation of first or second order. In a cartography scenario, it is crucial to project the three-dimensional earth onto a two-dimensional map. When applying VAEs, the problem is summarised to map from a two-dimensional Euclidean latent space (the map) to the three-dimensional observation space (the earth surface). If, by construction, the decoder exhibits regions of high curvature, the stage is especially bad for such methods.

We aim to improve the state of geodesics in deep latent-variable models with respect to runtime and use the geodesic as a distance metric. We expect that short (approximate) geodesics under the learned

model indicate similarity of data points in question. If we assume that a simple prior and a simple decoder are insufficient to represent complex data distributions and a complex decoder is detrimental to the calculation of geodesics, we need the prior to be sufficiently complex to allow the decoder to be sufficiently simple. Our solution is the use of a powerful hierarchical prior representation in the context of VAEs and the simple penalisation of the curvature of the decoder. We name this approach *flat manifold* variational autoencoder since the Riemannian manifold of the decoder is isometric to Euclidean space. We show empirically that the resulting model features geodesics which are approximated very well by the Euclidean distance. This effectively removes the need for numerical optimisation, reducing the calculation of an approximate geodesic to that of a simple Euclidean distance calculation in latent space. This is accompanied by a speedup of several orders of magnitude, rendering the method practical for applications in real-time scenarios.

## 2 VARIATIONAL AUTOENCODERS WITH RIEMANNIAN MANIFOLD REGULARISATION

### 2.1 BACKGROUND ON VAEs WITH HIERARCHICAL PRIORS

Latent-variable models (LVMs) are defined as

$$p(\mathbf{x}) = \int p(\mathbf{x}|\mathbf{z}) \, p(\mathbf{z}) \, \mathrm{d}\mathbf{z}, \tag{1}$$

where $\mathbf{z} \in \mathbb{R}^{N_z}$ represents latent variables and $\mathbf{x} \in \mathbb{R}^{N_x}$ the observable data. The integral in Eq. (1) is usually intractable but it can be approximated by maximising the evidence lower bound (ELBO) (Kingma & Welling, 2014; Rezende et al., 2014):

$$\mathbb{E}_{p_\mathcal{D}(\mathbf{x})} \big[ \log p_\theta(\mathbf{x}) \big] \geq \mathbb{E}_{p_\mathcal{D}(\mathbf{x})} \Big[ \mathbb{E}_{q_\phi(\mathbf{z}|\mathbf{x})} \big[ \log p_\theta(\mathbf{x}|\mathbf{z}) \big] - \mathbb{KL} \big( q_\phi(\mathbf{z}|\mathbf{x}) \| p(\mathbf{z}) \big) \Big], \tag{2}$$

where $p_\mathcal{D}(\mathbf{x}) = \frac{1}{N} \sum_{i=1}^{N} \delta(\mathbf{x} - \mathbf{x}_i)$ represents the empirical distribution of the data $\mathcal{D}$. The distribution parameters of the approximate posterior $q_\phi(\mathbf{z}|\mathbf{x})$ and the likelihood $p_\theta(\mathbf{x}|\mathbf{z})$ are represented by neural networks. The prior $p(\mathbf{z})$ is usually defined as the standard normal distribution. This model is commonly referred to the variational autoencoder (VAE).

However, a standard normal prior often leads to an over-regularisation of the approximate posterior, which results in a less informative learned latent representation of the data (Tomczak & Welling, 2018; Klushyn et al., 2019). To enable the model to learn an informative latent representation, Klushyn et al. (2019) propose to use a flexible hierarchical prior $p_\Theta(\mathbf{z}) = \int p_\Theta(\mathbf{z}|\zeta) \, p(\zeta) \, \mathrm{d}\zeta$, where $p(\zeta)$ is the standard normal distribution. Based on the insight that the optimal prior is the aggregated posterior (Tomczak & Welling, 2018), the above integral is approximated by an importance-weighted (IW) bound (Burda et al., 2015) using samples from $q_\phi(\mathbf{z}|\mathbf{x})$. This leads to a model with two stochastic layers and the following upper bound on the Kullback-Leibler ($\mathbb{KL}$) term:

$$\mathbb{E}_{p_\mathcal{D}(\mathbf{x})} \, \mathbb{KL} \big( q_\phi(\mathbf{z}|\mathbf{x}) \| p(\mathbf{z}) \big) \leq \mathcal{F}(\phi, \Theta, \Phi)$$

$$\equiv \mathbb{E}_{p_\mathcal{D}(\mathbf{x})} \, \mathbb{E}_{q_\phi(\mathbf{z}|\mathbf{x})} \left[ \log q_\phi(\mathbf{z}|\mathbf{x}) - \mathbb{E}_{\zeta_{1:K} \sim q_\Phi(\zeta|\mathbf{z})} \left[ \log \frac{1}{K} \sum_{i=1}^{K} \frac{p_\Theta(\mathbf{z}|\zeta_i) \, p(\zeta_i)}{q_\Phi(\zeta_i|\mathbf{z})} \right] \right], \tag{3}$$

where $K$ is the number of importance samples. Since it has been shown that high ELBO values do not necessarily correlate with informative latent representations (Alemi et al., 2018; Higgins et al., 2017)—which is also the case for hierarchical models (Sønderby et al., 2016)—different optimisation approaches have been introduced (Bowman et al., 2016; Sønderby et al., 2016). Klushyn et al. (2019) follow the line of argument in (Rezende & Viola, 2018) and reformulate the resulting ELBO as the Lagrangian of a constrained optimisation problem:

$$\mathcal{L}_{\text{VHP}}(\theta, \phi, \Theta, \Phi; \lambda) \equiv \mathcal{F}(\phi, \Theta, \Phi) + \lambda \big( \mathbb{E}_{p_\mathcal{D}(\mathbf{x})} \, \mathbb{E}_{q_\phi(\mathbf{z}|\mathbf{x})} \big[ \mathrm{C}_\theta(\mathbf{x}, \mathbf{z}) \big] - \kappa^2 \big), \tag{4}$$

with the optimisation objective $\mathcal{F}(\phi, \Theta, \Phi)$, the inequality constraint $\mathbb{E}_{p_\mathcal{D}(\mathbf{x})} \, \mathbb{E}_{q_\phi(\mathbf{z}|\mathbf{x})} \big[ \mathrm{C}_\theta(\mathbf{x}, \mathbf{z}) \big] \leq \kappa^2$, and the Lagrange multiplier $\lambda$. $\mathrm{C}_\theta(\mathbf{x}, \mathbf{z})$ is defined as the reconstruction-error-related term in $-\log p_\theta(\mathbf{x}|\mathbf{z})$. Thus, we obtain the following optimisation problem:

$$\min_{\Theta, \Phi} \min_{\theta} \max_{\lambda} \min_{\phi} \mathcal{L}_{\text{VHP}}(\theta, \phi, \Theta, \Phi; \lambda) \quad \text{s.t.} \quad \lambda \geq 0. \tag{5}$$

Building on that, the authors propose an optimisation algorithm—including a $\lambda$-update scheme—to achieve a tight lower bound on the log likelihood. This approach is referred to as variational hierarchical prior VHP-VAE.

## 2.2 Background on Riemannian Geometry for VAEs

A manifold $M$ is a space which is differentiable and locally Euclidean. Given $M$, a Riemannian manifold is $(M, \mathbf{G})$, where $\mathbf{G} \in \mathbb{R}^{N_z \times N_z}$ is the Riemannian metric tensor. At each point $\mathbf{z} \in M$ in the latent space, the corresponding metric tensor $\mathbf{G}$ defines an inner product in the tangent space $\mathbf{z}' \in T_{\mathbf{z}}M$:

$$\langle \mathbf{z}', \mathbf{z}' \rangle_{\mathbf{z}} \equiv \mathbf{z}'^{T}\, \mathbf{G}(\mathbf{z})\, \mathbf{z}'. \tag{6}$$

Chen et al. (2018a); Arvanitidis et al. (2018) define the latent space of a VAE as a Riemannian manifold, which allows for computing the *observation space distance* based on distances in the latent space. Given a smooth trajectory $\gamma : [0, 1] \to \mathbb{R}^{N_z}$ in the Riemannian (latent) space and the corresponding $N_x$-dimensional Euclidean (observation) space, the Riemaninan distance of the curve can be written as

$$L(\gamma) = \int_0^1 \phi(t)\, \mathrm{d}t, \qquad \phi(t) \equiv \sqrt{\langle \gamma'(t), \gamma'(t) \rangle_{\gamma(t)}} = \sqrt{\gamma'(t)^T \mathbf{G}(\gamma(t))\gamma'(t)}, \tag{7}$$

where $\mathbf{G}$ depends on $\gamma$, $\phi(t)$ denotes the Riemannian velocity and $\gamma'(t)$ represents the time-derivative of the trajectory. In the VAE models, $\gamma$ is transformed by a continuous function $f(\gamma(t))$ (decoder) to $\mathbf{x}$, and the metric tensor is defined as $\mathbf{G} = \mathbf{J}^T\mathbf{J}$, where $\mathbf{J}$ is the Jacobian of the decoder. The geodesic is obtained by minimising $L(\gamma)$.

The magnification factor ($\mathrm{MF}(\mathbf{z}) \equiv \sqrt{\det \mathbf{G}(\mathbf{z})}$) (Bishop et al., 1997) shows the sensitivity of the likelihood functions. When projecting from the Riemannian (latent) to the Euclidean (observation) space, the MF can be considered a scaling coefficient.

## 2.3 Flat Manifold VAE

In this work, we aim to compute geodesics directly in the latent space by measuring the Euclidean distance between encoded data points. For this purpose, the metric tensor, which describes our latent space, needs to be $\mathbf{G} \propto \mathbb{1}$—hence a Euclidean metric. This simplifies the computation of geodesics (Eq. (7)) to

$$L(\gamma) = \int_0^1 \phi(t)\, \mathrm{d}t \propto \|\mathbf{z}(1) - \mathbf{z}(0)\|. \tag{8}$$

We refer to a manifold $M$ with this property as *flat manifold* (Lee, 2006). As a consequence, our model must be capable of learning such *flat manifold* latent spaces, which typically requires complex latent representations of the data (see experiments in Sec. 4). Therefore, we propose the following approach: (i) to enable our model to learn complex latent representations, we introduce a flexible prior, which is learned by the model (empirical Bayes); and (ii) we penalise the curvature of the decoder such that $\mathbf{G} \propto \mathbb{1}$.

For this purpose, we extend the VHP-VAE introduced in Sec. 2.1 by a Jacobian-regularisation term. Defining the regularisation term as part of the constraint is in line with the constrained optimisation setting. The resulting objective function is

$$\mathcal{L} = \mathcal{F}(\phi, \Theta, \Phi) + \lambda\big( \mathbb{E}_{p_{\mathcal{D}}(\mathbf{x})} \mathbb{E}_{q_{\phi}(\mathbf{z}|\mathbf{x})} \big[\mathrm{C}_{\theta}(\mathbf{x}, \mathbf{z}) + \beta \big\| \mathbf{J}(\mathbf{z})^T \mathbf{J}(\mathbf{z}) - c^2 \mathbb{1} \big\| \big] - \kappa^2 \big), \tag{9}$$

where $\beta$ is a hyper-parameter determining the influence of the regularisation. $c^2$ is defined to be the mean over the batch samples and diagonal elements of $\mathbf{J}^T\mathbf{J}$, which we view as a normalisation process. Additionally, we use a stochastic approximation (first order Taylor expansion) of the Jacobian (Rifai et al., 2011b) to improve the computational efficiency:

$$\mathbf{J}(\mathbf{z}) = \lim_{\sigma \to 0} \frac{1}{\sigma} \mathbb{E}_{p_{\mathcal{D}}(\mathbf{x})} \mathbb{E}_{q_{\phi}(\mathbf{z}|\mathbf{x})} [f(\mathbf{z} + \epsilon) - f(\mathbf{z})], \tag{10}$$

where $\epsilon \sim \mathcal{N}(0, \sigma^2 I)$. This approximation method allows for a faster computation of the gradient and avoids the second-derivative problem of piece-wise linear layers (Chen et al., 2018a) during optimisation.

However, the regularisation term in Eq. (9) only effects the decoder function in regions where data is available. To overcome this issue, we propose to use *mixup*, a data-augmentation method (Zhang et al., 2018), which was introduced in the context of supervised learning. We extend this method to the VAE framework (unsupervised learning) by applying it to encoded data in the latent space. Our aim is to augment data by interpolating between two encoded data points $\mathbf{z}_i$ and $\mathbf{z}_j$:

$$g_{\text{aug}}(\mathbf{z}_i, \mathbf{z}_j) = (1 - \alpha)\,\mathbf{z}_i + \alpha\,\mathbf{z}_j, \tag{11}$$

with $\mathbf{x}_i, \mathbf{x}_j \sim p_{\mathcal{D}}(\mathbf{x})$, $\mathbf{z}_i \sim q_\phi(\mathbf{z}|\mathbf{x}_i)$, $\mathbf{z}_j \sim q_\phi(\mathbf{z}|\mathbf{x}_j)$, and $\alpha \sim U(-\alpha_0, 1 + \alpha_0)$. In contrast to (Zhang et al., 2018), where $\alpha \in [0, 1]$ limits the data augmentation to only convex combinations, we define $\alpha_0 > 0$ to take into account the outer edge of the data manifold. We obtain the objective function of our *flat manifold* VAE (FMVAE) by combining *mixup* (Eq. (11)) with Eq. (9):

$$\mathcal{L}_{\text{VHP-FMVAE}} = \mathcal{L}_{\text{VHP}} + \lambda\,\beta\,\mathbb{E}_{\mathbf{x}_i, \mathbf{x}_j \sim p_{\mathcal{D}}(\mathbf{x})}\,\mathbb{E}_{\mathbf{z}_i \sim q_\phi(\mathbf{z}|\mathbf{x}_i), \mathbf{z}_j \sim q_\phi(\mathbf{z}|\mathbf{x}_j)}\left[\left\|\mathbf{G}(g_{\text{aug}}(\mathbf{z}_i, \mathbf{z}_j)) - c^2\mathbb{1}\right\|\right]. \tag{12}$$

By using augmented data when minimising $\|\mathbf{G} - c^2\mathbb{1}\|$, we regularise $\mathbf{G}$ to be a scaled identity matrix for the *entire* latent space enclosed by our data manifold. Hence, our VAE learns a scaled Euclidean latent space, where $c$ is the scale factor. Therefore, the function $f(\mathbf{z})$ (decoder) is—up to the scale factor $c$—isometry/distance-preserving since $D_{\mathbf{x}}(f(\mathbf{z}_i), f(\mathbf{z}_j)) \approx c\,D_{\mathbf{z}}(\mathbf{z}_i, \mathbf{z}_j)$, where $D$ refers to the distance between two data points in the observation and latent space, respectively.

The decoder of the proposed approach satisfies the Lipschitz continuity condition. Given the Lipschitz continuity condition $D_{\mathbf{x}}(f(\mathbf{z}_i), f(\mathbf{z}_j)) \leq a\,D_{\mathbf{z}}(\mathbf{z}_i, \mathbf{z}_j)$, where $a$ is the Lipschitz constant, we consider the decoder function, and hence the latent space as *smooth* if $\exists\, c \leq a$.

## 3 RELATED WORK

**Latent space of VAEs**. In general, the latent space of VAEs is considered to be Euclidean (e.g. Kingma et al., 2016; Higgins et al., 2017), but they are not constrained to be Euclidean. This can be problematic if a precise metric is required in the latent space. Some recent works (Mathieu et al., 2019; Grattarola et al., 2018) adapted the latent space to be non-Euclidean to match the data structure. We solve the problem from another perspective by enforcing the latent space to be Euclidean.

**Jacobian and Hessian regularisation**. In (Rifai et al., 2011a), the authors propose to regularise the Jacobian and Hessian of the encoder. However, the encoders of VAEs are already regularised by the $\mathbb{KL}$ term. Furthermore, it is more difficult augment data in the observation space than in the latent space for data augmentation. Encoder regularisation enables the model to perform better in case of, e.g., object recognition from the latent space. By contrast, decoder regularisation enables the model to do tasks such as generating motions based on the latent space. In (Hadjeres et al., 2017), the Jacobian of the decoder was regularised to be as small as possible/zero. By contrast, we regularise the Jacobian to be constand, and hence the Hessian to be zero leading to a correct metric in the latent space. Nie & Patel (2019) regularised the Jacobian with respect to the weights of both the encoder and decoder for GANs. In terms of supervised learning, Jakubovitz & Giryes (2018) regularised the Jacobian to improve the robustness for classification.

**Metric learning**. Various metric learning approaches for both deep supervised and unsupervised models were proposed. For instance, deep metric learning (Hoffer & Ailon, 2015) used a triplet network for supervised learning. Karaletsos et al. (2016) introduced an unsupervised metric learning method, where a VAE is combined with triplets. However, a human oracle is still required. By contrast, our approach is completely based on unsupervised learning, using the tangent space of the decoder as a distance metric. Our proposed method is similar to the metric learning methods such as Large Margin Nearest Neighbor (Weinberger & Saul, 2009), which pulls target neighbours together and pushes imposters away, but our approach is an unsupervised method.

**Constraints in latent space**. Constraints on time (e.g. Wang et al., 2007; Chen et al., 2016; 2015) allow to obtain similar distance metrics in the latent space. However, our method can be used for general datasets without sequential data. Additionally, constraints on time cannot guarantee that

the metric is correct in between of different time steps. By contrast, in case of sequential data, our method can be used to obtain a correct metric through data augmentation.

**Data augmentation**. In regions without training data, the latent space is trained arbitrarily. Our method is able to augment data in the latent space, so that we can smoothly interpolate between two points even in case the data in between is missing in the training dataset using mixup. Various follow-up studies of *mixup* were developed, such as (Verma et al., 2018; Beckham et al., 2019). GANs which generates fake data in the latent space are a similar approach as our approach.

**Geodesic**. There have been some recent studies on geodesics for generative models using both stochastic methods (e.g. Tosi et al., 2014; Arvanitidis et al., 2018; Hauberg, 2018; Chen et al., 2018b) and deterministic approaches (e.g., Chen et al., 2018a; 2019). The main difference is that the stochastic methods work for the regions without data, because the RBF layer generates high MF for those—however, it is less general. The uncertainty does not emerge from a principled way (such as in a Bayesian model) but is instead driven by certain assumptions. The deterministic method requires other strategies to guarantee that the geodesic is within the data manifold. In our proposed method, we regularise the latent space to have a similar metric, so that we do not need to consider low density regions. In previous work, methods were introduced for computing/finding the geodesic in the latent space. However, it is a novel approach to use the geodesic/Riemannian distance for influencing the latent representation. Tenenbaum et al. (2000) projected the latent space to a new latent space where the geodesic is equivalent to the Euclidean interpolation. However, the two separate processes—VAEs and projection—probably cannot allow the model to find the latent features autonomously.

## 4 EXPERIMENTS

We test our method on artificial pendulum images, human motion, the MNIST and the MOT16 datasets. We measure the performance in terms of equidistances, interpolation smoothness and geodesics. Additionally, our method is applied to a real-world environment—a tracking system from the context of autonomous driving. Consequently, the tracking and re-identification capabilities are evaluated.

Riemannian metric tensor has many intrinsic properties of a manifold and measures local angles, length, surface area and volumes (Bronstein et al., 2017). Therefore, the models are quantified using the Riemannian metric tensor—condition numbers and MFs. The condition number which shows the ratio of the most elongated to the least elongated direction is defined as $k(\mathbf{G}) = \frac{S_{\max}(\mathbf{G})}{S_{\min}(\mathbf{G})}$, where $S$ is an eigenvalue of $\mathbf{G}$. Since we cannot directly compare the MFs of different models, the MFs are normalised through dividing by their mean. Accordingly, we can measure how the MFs spread out from their mean. The model is more invariant with respect to the metric tensor if the condition number is smaller and the normalised MF is closer to one.

### 4.1 PENDULUM IMAGE DATASET

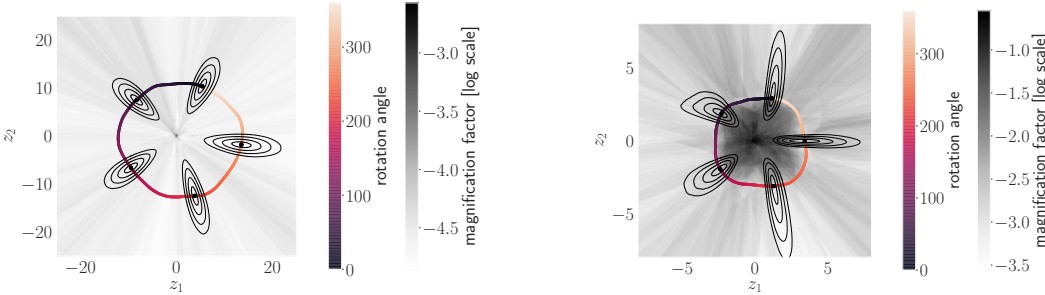

(a) Latent representation of VHP-FMVAE.          (b) Latent representation of VHP-VAE.

Figure 1: Equidistance in the latent space of the pendulum dataset. The black curves are points of equal distance to a center. The distance is computed using (7).

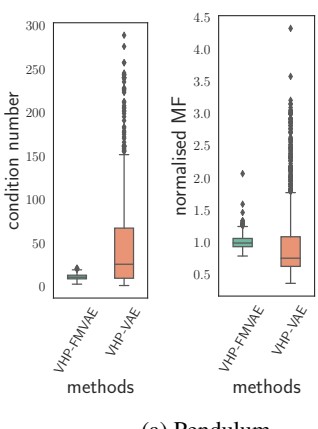
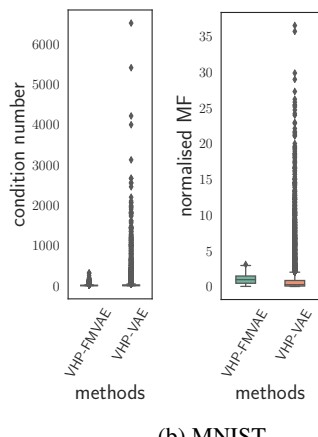

(a) Pendulum.                    (b) MNIST.

Figure 2: Boxplot of the condition number and the normalised MF for pendulum and MNIST datasets. The pendulum and MNIST have 10,000 and 1,000 samples generated in from the latent space, respectively.

The pendulum dataset (Klushyn et al., 2019; Chen et al., 2018a) consists of $16 \times 16$-pixel images generated by a pendulum simulator. We generated $15 \cdot 10^3$ images with joint angle in the ranges of $[0, 360)$ degrees. Additionally, we added 0.05 Gaussian noise to each pixel to avoid overfitting.

Fig. 1 shows the equidistance plots for five different encoded data points. VHP-FMVAE smoothens the MF, while VHP-VAE has large area of high MF in the middle. Without regularisation, the contour of the equidistances are significant different from high MF areas to low MF areas. Fig. 2a shows that the condition number of the VHP-FMVAE is smaller than that of the VAE-VHP in terms of the Riemannian tensor. Additionally, the normalised MF of the VHP-FMVAE is closer to one.

In order to avoid bias visualisation in Fig. 1, 3 and 7, we reset the range of the MF for plotting while not changing the MF values. In Fig. 1a, Fig. 3a and 7, the upper range is set to be $\frac{\max(\mathrm{MF}_2(\mathrm{grid\_area})) \cdot \mathrm{mean}(\mathrm{MF}_1(\mathrm{data}))}{\mathrm{mean}(\mathrm{MF}_2(\mathrm{data}))}$. $\mathrm{MF}_1$ and $\mathrm{MF}_2$ refer to the MF of VHP-FMVAE with/without Jacobian normalisation/*mixup* and VAE-VHP, respectively. $\mathrm{MF}(\mathrm{data})$ and $\mathrm{MF}(\mathrm{grid\_area})$ are the MF of the training data and the MF of the grid area, respectively.

## 4.2 HUMAN MOTION

To evaluate our approach, CMU human motion dataset (`http://mocap.cs.cmu.edu`) is used. Walking (subject 35), jogging (subject 35), balancing (subject 49), punching (subject 143) and kicking (subject 74) are selected for the experiment. After data pre-processing, the input data is a 50-dimensional vector of the joint angles. The dataset is unbalanced—walking dataset size is larger than that of jogging.

Table 1: The length ratio of Euclidean interpolation to geodesic. The Riemannian distance and the distance in the latent space are computed. We randomly sample 100 pairs of points and interpolate between each pair. The mean and the standard deviation of the ratios are listed below.

| DATASET | METHOD | RIEMANNIAN | LATENT |
|---------|--------|------------|--------|
| HUMAN | VHP-FMVAE | **1.02 ± 0.06** | **0.93 ± 0.03** |
| | VHP-VAE | 1.23 ± 0.20 | 0.82 ± 0.10 |
| MNIST | VHP-FMVAE | **1.01 ± 0.08** | **0.92 ± 0.05** |
| | VHP-VAE | 1.13 ± 0.22 | 0.70 ± 0.31 |

**Equidistance**. We randomly select a point from each class as the centre of the equidistance. As shown in Fig. 3, the proposed method has more similar equidistance at different locations in the latent space, while in the model without regularisation, the equidistance contour is distorted in the high MF

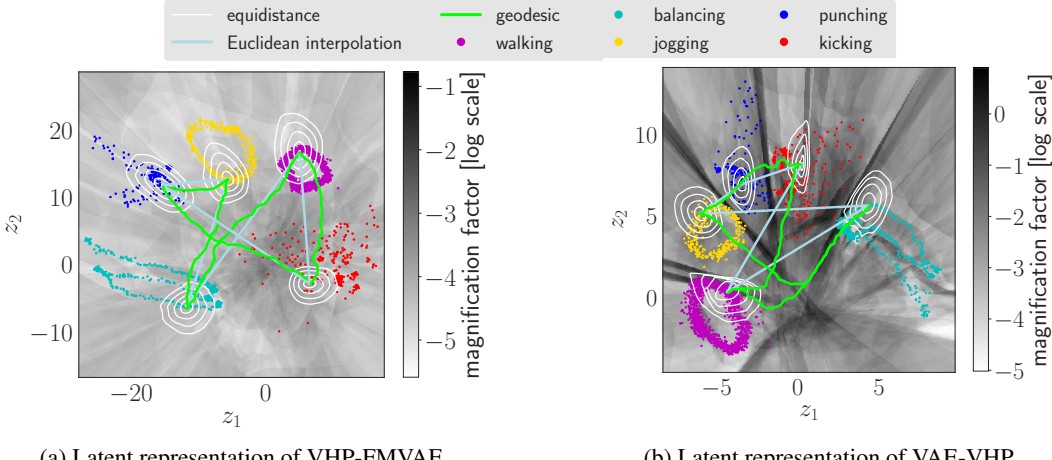

(a) Latent representation of VHP-FMVAE.  (b) Latent representation of VAE-VHP.

Figure 3: Equidistance in the latent space of the human motion dataset. (a) Jogging is a large-range movement compared with walking, so that jogging is reasonably distributed on a larger area in the latent space than that of walking. (b) In contrast, without regularisation, walking is larger than the jogging in the latent space. For FMVAE, the Euclidean interpolatioins are much closer to the geodesics.

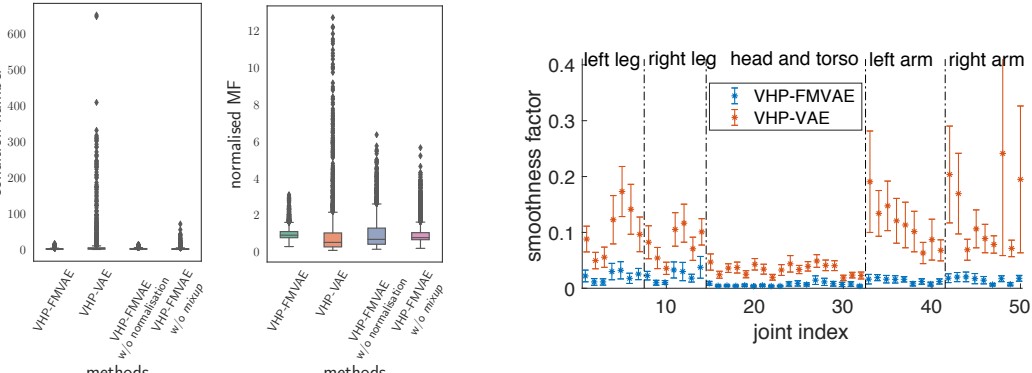

(a) Condition number.  (b) Normalised MF.

Figure 4: Boxplot of the condition number and the normalised MF of human motion dataset.

Figure 5: Smoothness of the human dataset. The mean and standard deviation are shown. The smaller the value is, the smoother the model is.

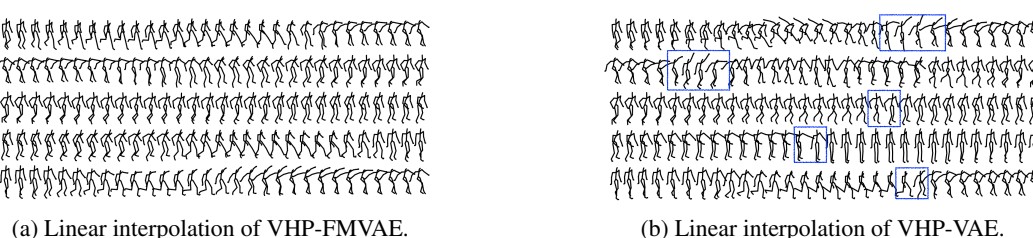

(a) Linear interpolation of VHP-FMVAE.  (b) Linear interpolation of VHP-VAE.

Figure 6: Generated movements of the human motion dataset. The abrupt motions are marked by blue boxes.

area. The latent space of VHP-FMVAE reflects the true distribution of the data (see more details in Fig. 3). Moreover, we compute the Riemannian tensors for 3,000 samples randomly generated from the latent space. In different locations of a latent space, the Riemannian tensors of VHP-FMVAE is more invariant than that of VAE-VHP (see Fig. 4).

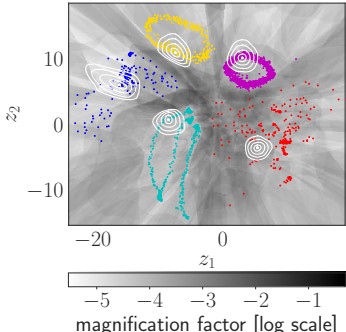

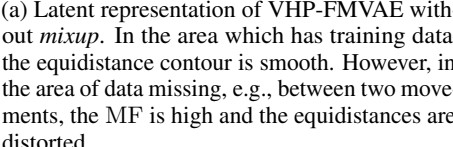

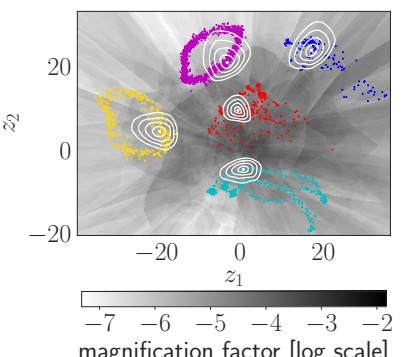

(a) Latent representation of VHP-FMVAE without *mixup*. In the area which has training data, the equidistance contour is smooth. However, in the area of data missing, e.g., between two movements, the MF is high and the equidistances are distorted.

(b) Latent representation of VHP-FMVAE without the Jacobian normalisation. Although it does not have extreme sharp equidistance contour, the equidistance is still scaled in various locations of the latent space. Additionally, the distribution of walking in the latent space is still larger than that of jogging.

Figure 7: Influence of the data augmentation and the Jacobian normalisation. The movements are coloured the same as Fig. 3.

**Smoothness**. We randomly sample 100 pair points and linearly interpolate between each pair. The second derivative of each trajectory is defined as the smoothness factor. Fig. 5 illustrates that VHP-FMVAE significantly outperforms the latent space of VAE-VHP in terms of the smoothness. Fig. 6 shows five examples of the interpolated trajectories.

**Geodesic**. We compare the proposed method with the graph-based geodesic approach (Chen et al., 2019) which approximates the geodesic using a graph in a generative model. The graph-based approach is much faster than previous geodesic search method such as (Chen et al., 2018a). The graph of the baseline has 14,400 nodes which are sampled in the latent space using uniform distribution. Each node has 12 neighbours. The regularisation of singular value decomposition (SVD) (Chen et al., 2018a) is 0.001 for VAE-VHP while it is adapted to VHP-FMVAE based the mean value of the square of the singular.

Table 1 shows the ratios from Euclidean interpolations to geodesics. If the ratio of the distance is close to one, the Euclidean interpolation is able to approximate the geodesic. Table 1 demonstrates that the Euclidean interpolation of VHP-FMVAE is more close to geodesic, compared with VAE-VHP. Additionally, the proposed method is 1,000 times faster than the graph-based method in terms of searching for the geodesics. Fig. 3 depicts five examples of the geodesics.

**Influence of the data augmentation and the Jacobian normalisation**. Fig. 7a shows the influence of the data augmentation. The samples of the regularisation term are the same as $\mathcal{L}_{\text{VHP}}$. Fig. 7b illustrates the influence of the Jacobian normalisation. We removed the normalisation, and consequently the regularisation term is $\|\mathbf{G}(g_{\text{aug}}(\mathbf{z}_i, \mathbf{z}_j))\|$. The $c^2 \mathbb{1}$ term in the regularisation is necessary; otherwise it only has dissimilarity constraints, but cannot reduce the distance for points with high similarities. For instance, the walking is not squeezed in the latent space (see Fig. 4). By contrast, regularising the Jacobian to be constant elongates the distance in the latent space with high MF areas while squeezing the distance with low MF areas.

### 4.3 MNIST

A fixed binarised version of the MNIST digit dataset (Larochelle & Murray, 2011) is used to evaluate our approach. The dataset consists of 50,000 training and 10,000 test images of handwritten digits (zero to nine) with $28 \times 28$ pixels in size.

The equidistances of VHP-FMVAE are more invariant and smoother than that of VAE-VHP (see Fig. 8 and Fig. 4). Similar as the human motion dataset, the geodesic of our method is more similar to Euclidean interpolation, compared with VAE-VHP, which indicates that the latent space of the VHP-FMVAE is able to approximate geodesic (see Table 1).

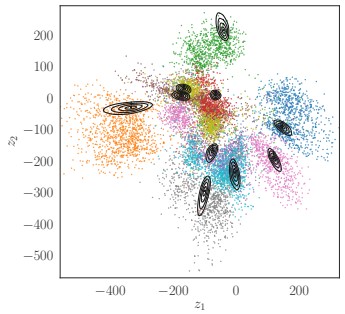

(a) Latent representation of VHP-FMVAE.

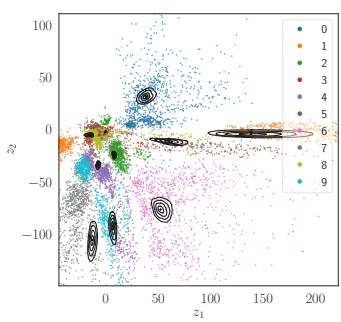

(b) Latent representation of VHP-VAE.

Figure 8: Equidistance in the latent space of MNIST dataset. (b) The data ranges on $z_1$ and $z_2$ of the VHP-VAE are [-106.21, 369.38] and [-365.64, 164.08], respectively. For better visualisation, we crop out the less dense areas.

## 4.4 TRACKING

We evaluate our approach on the MOT16 object-tracking database (Milan et al., 2016), which is a large-scale person re-identification dataset, containing both static and dynamic scenes from diverse cameras.

Table 2: Comparisons between different descriptors for the purposes of object tracking and re-identification (Ristani et al., 2016). The bold and the red numbers denote the best results among all methods and among non-supervised methods, respectively.

| METHOD | TYPE | IDF$_1$↑ | IDP↑ | IDR↑ | RECALL↑ | PRECISION↑ | FAR↓ | MT↑ |
|---|---|---|---|---|---|---|---|---|
| VHP-FMVAE-SORT $\beta = 300$ (OURS) | UNSUPERVISED | 63.7 | 77.0 | 54.3 | 65.0 | 92.3 | **1.12** | 158 |
| VHP-FMVAE-SORT $\beta = 3000$ (OURS) | UNSUPERVISED | 64.2 | **77.6** | 54.8 | 65.1 | **92.3** | 1.13 | 162 |
| VHP-VAE-SORT | UNSUPERVISED | 60.5 | 72.3 | 52.1 | 65.8 | 91.4 | 1.28 | 170 |
| SORT | N.A. | 57.0 | 67.4 | 49.4 | 66.4 | 90.6 | 1.44 | 158 |
| DEEPSORT | SUPERVISED | **64.7** | 76.9 | **55.8** | **66.7** | 91.9 | 1.22 | **180** |

| METHOD | PT↓ | ML↓ | FP↓ | FN↓ | IDs↓ | FM↓ | MOTA ↑ | MOTP ↑ | MOTAL↑ |
|---|---|---|---|---|---|---|---|---|---|
| VHP-FMVAE-SORT $\beta = 300$ (OURS) | 269 | 90 | **5950** | 38592 | 616 | **1143** | 59.1 | 81.8 | 59.7 |
| VHP-FMVAE-SORT $\beta = 3000$ (OURS) | 265 | 90 | 6026 | 38515 | 598 | 1163 | 59.1 | 81.8 | 59.7 |
| VHP-VAE-SORT | 266 | **81** | 6820 | 37739 | 693 | 1264 | 59.0 | 81.6 | 59.6 |
| SORT | 275 | 84 | 7643 | 37071 | 1486 | 1515 | 58.2 | **81.9** | 59.5 |
| DEEPSORT | **250** | 87 | 6506 | **36747** | **585** | 1165 | **60.3** | 81.6 | **60.8** |

We compare with two baselines: SORT (Bewley et al., 2016) and DeepSORT (Wojke et al., 2017). SORT is a simple online and realtime tracking method, which uses bounding box intersection-over-union (IOU) for associating detections between frames and Kalman filters for the track predictions. It relies on good two-dimensional bounding box detections from a separate detector, and suffers from ID switching when tracks overlap in the image. DeepSORT extends the original SORT algorithm to integrate appearance information based on a deep appearance descriptor, which helps with re-identification in the case of such overlaps or missed detections. The deep appearance descriptor is trained using a *supervised* cosine metric learning approach (Wojke & Bewley, 2018). The candidate object locations of the pre-generated detections for both SORT, DeepSORT and our method are taken from (Yu et al., 2016). Further details regarding the implementation can be found in App. A.3.

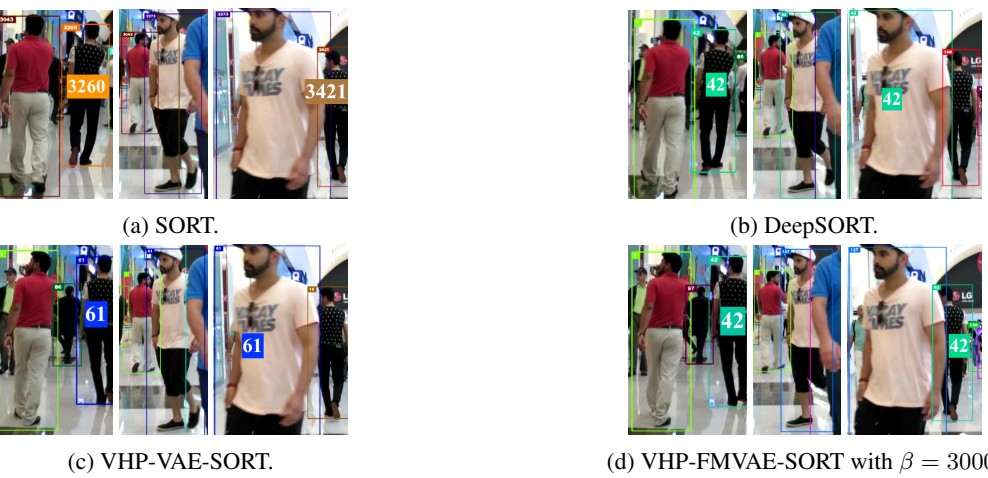

(a) SORT.

(b) DeepSORT.

(c) VHP-VAE-SORT.

(d) VHP-FMVAE-SORT with $\beta = 3000$.

Figure 9: Example identity switches between overlapping tracks. For vanilla SORT, track 3260 gets occluded and when subsequently visible, it gets assigned a new ID 3421. For deeSORT and VHP-VAE-SORT, the occluding track gets assigned the same ID as the track it occludes (42/61), and subsequently keeps this (erroneous) track. For VHP-FMVAE-SORT, the track 42 gets occluded, but is re-identified correctly when again visible.

We use the following metrics for evaluation. $\uparrow$ indicates that the higher the score is, the better the performance is. On the contrary, $\downarrow$ indicates that the lower the score is, the better the performance is.

· IDF$_1$($\uparrow$): ID F$_1$ Score
· IDP($\uparrow$): ID Precision
· IDR($\uparrow$): ID Recall
· FAR($\downarrow$): False Alarm Ratio
· MT($\uparrow$): Mostly Tracked Trajectory
· PT($\downarrow$): Partially Tracked Trajectory
· ML($\downarrow$): Mostly Lost Trajectory
· FP($\downarrow$): False Positives

· FN($\downarrow$): False Negatives
· IDs($\downarrow$): Number of times an ID switches to a different previously tracked object
· FM($\downarrow$): Fragmentations
· MOTA($\uparrow$): Multi-object tracking accuracy
· MOTP($\uparrow$): Multi-object tracking precision
· MOTAL($\uparrow$): Log tracking accuracy

Table 2 shows that the performance of the proposed method is better than that of the model without Jacobian regularisation, and even close to the the performance of supervised learning. All methods depend on the same underlying detector for object candidates, and identical Kalmann filter parameters. Compared to baseline SORT which does not utilise any appearance information, DeepSORT has 2.54 times, VHP-VAE-SORT has 2.14 times, VHP-FMVAE-SORT ($\beta = 300$) has 2.41 times and VHP-FMVAE-SORT ($\beta = 3000$) has 2.48 times fewer ID switches. Whilst the supervised DeepSORT descriptor has the least, using unsupervised VAEs with flat decoders has only 2.2% more switches, without the need for labels. Furthermore, by ensuring a quasi-Euclidean latent space, one can query nearest-neighbours efficiently via data-structures such as kDTrees. Fig. 9 shows an example of the results. In other examples of the videos, the VHP-FMVAE-SORT works similar as the deepSORT. Videos of the results can be downloaded at: `http://tiny.cc/0s7lcz`

## 5 CONCLUSION

In this paper, we have proposed a novel approach, which we call *flat manifold* variational autoencoder. We have shown that—using this method—geodesics can be computed directly in the latent space by measuring the Euclidean distance between encoded data points. This is realised by combining a powerful empirical Bayes prior with a Jacobian-regularisation method that constrains the learned latent space to be Euclidean. Consequently, geodesic can be approximated 1,000 times faster than comparable state-of-the-art methods. Furthermore, using the approximated geodesic as a distance function, we have evaluated our approach on the MOT16 object-tracking database showing comparable performance as in case of supervised learning.

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

# A   APPENDIX

## A.1   VECTOR FIELD

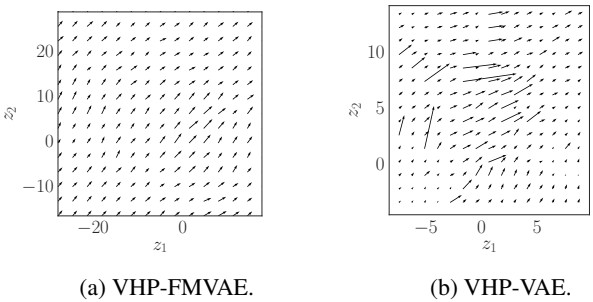

(a) VHP-FMVAE.                    (b) VHP-VAE.

Figure 10: Vector field of the human motion dataset. The vector field is a vector of $L_2$ norm over the output of Jacobian. The figures are corresponding to Fig. 3. The vector field of VHP-FMVAE is more regular than that of VAE-VHP.

## A.2   MODEL ARCHITECTURES

Table 3: Model architectures. FC refers to fully-connected layers. Conv2D and Conv2DT denote tow-D convolution layer and transposed two-D convolution layer, respectively. See the definition of $\nu$ in (Klushyn et al., 2019).

| DATASET | OPTIMISER | ARCHITECTURE | |
| --- | --- | --- | --- |
| PENDULUM | ADAM 1e-4 | INPUT | $16\times16\times1$ |
| | | LATENTS | 2 |
| | | $q_\phi(\mathbf{z}|\mathbf{x})$ | FC 256, 256. ReLU activation. |
| | | $p_\theta(\mathbf{x}|\mathbf{z})$ | FC 256, 256. ReLU activation. Gaussian. |
| | | $q_\Phi(\zeta|\mathbf{z})$ | FC 256, 256, ReLU activation. |
| | | $p_\Theta(\mathbf{z}|\zeta)$ | FC 256, 256, ReLU activation. |
| | | OTHERS | $\kappa = 0.025$, $\nu = 1$, $K = 16$, $\beta = 1000$. |
| CMU HUMAN | ADAM 1e-4 | INPUT | 50 |
| | | LATENTS | 2 |
| | | $q_\phi(\mathbf{z}|\mathbf{x})$ | FC 256, 256, 256, 256. ReLU activation. |
| | | $p_\theta(\mathbf{x}|\mathbf{z})$ | FC 256, 256, 256, 256. ReLU activation. Gaussian. |
| | | $q_\Phi(\zeta|\mathbf{z})$ | FC 256, 256, 256, 256, ReLU activation. |
| | | $p_\Theta(\mathbf{z}|\zeta)$ | FC 256, 256, 256, 256, ReLU activation. |
| | | OTHERS | $\kappa = 0.03$, $\nu = 1$, $K = 32$, $\beta = 8000$. |
| MNIST | ADAM 1e-4 | INPUT | $28\times28\times1$ |
| | | LATENTS | 2 |
| | | $q_\phi(\mathbf{z}|\mathbf{x})$ | FC 256, 256, 256, 256. ReLU activation. |
| | | $p_\theta(\mathbf{x}|\mathbf{z})$ | FC 256, 256, 256, 256. ReLU activation. Bernoulli. |
| | | $q_\Phi(\zeta|\mathbf{z})$ | FC 256, 256, 256, 256. ReLU activation. |
| | | $p_\Theta(\mathbf{z}|\zeta)$ | FC 256, 256, 256, 256. ReLU activation. |
| | | OTHERS | $\kappa = 0.245$, $\nu = 1$, $K = 16$, $\beta = 8000$. |
| MOT16 | ADAM 3e-5 | INPUT | $64\times64\times3$ |
| | | LATENTS | 128 |
| | | $q_\phi(\mathbf{z}|\mathbf{x})$ | VGG16 (Simonyan & Zisserman, 2015) |
| | | $p_\theta(\mathbf{x}|\mathbf{z})$ | Conv2DT+Conv2D 256, 128, 64, 32, 16. ReLU activation. Gaussian. |
| | | $q_\Phi(\zeta|\mathbf{z})$ | FC 512, 512. ReLU activation. |
| | | $p_\Theta(\mathbf{z}|\zeta)$ | FC 512, 512. ReLU activation. |
| | | OTHERS | $\kappa = 0.8$, $\nu = 1$, $K = 8$, $\beta = 300$ or 3000. |

### A.3 IMPLEMENTATION OF VHP-FMVAE-SORT

We evaluate the performance of our model by replacing the appearance descriptor from DeepSORT with the latent space embedding from the various auto-encoders used, using the same size of 128. The hyperparameters used were held constant: the minimum detection confidence of $0.3$, NMS max overlap of $0.7$, max cosine distance $0.2$, max appearance budget $100$. We tested a VHP-FMVAE, and our regularised VHP-FMVAE with $\beta = 300$ and $\beta = 3000$.

