# OpenReview forum: "FLAT MANIFOLD VAES"
_ICLR.cc/2020/Conference — Reject_

### Official Review · AnonReviewer1 · 2019-10-09
**Official Blind Review #1**

**Rating:** 6

**Review:**

Notes:

  -This paper suggests the use of VAEs with stronger priors along with more powerful regularization of the decoder (especially its curvature).  This lowering of curvature is seen as "flattening".

  -Paper uses the normal VAE ELBO.

  -The normal prior over-regularizes the approximate posterior.  One proposal is to use a "hierarchical prior": integral(p(z|zeta)*p(zeta), zeta) where zeta is a normal distribution.  So basically a function can transform the prior.

  -Importance weighting with q(z|x) has been proposed as a way to define a valid learning objective for this setting.

  -Another objective using lagrangian is called "VHP-VAE".

  -This paper extends VHP-VAE with jacobian regularization, which is approximated (the paper doesn't say so but I think it's a first order taylor expansion).

  -Paper also uses mixup in the latent space to provide regularization at points farther from the data.

  -With this mixup objective the mixing is also done to consider extrapolations in addition to interpolations.

  -The resulting latent space does indeed look much better (Figure 1).

  -The condition number is also way better (2a, 2b).

  -In figure 2, the background color indicates the degree of magnification (so the VAE-VHP has greater variability in distances?)  I found this figure a bit hard to itnerpret.

Comments:

  -This paper cites Mixup but there are two more papers to consider here: Manifold Mixup (ICML 2019) and Adversarial Mixup Resynthesis (Neurips 2019) which both considered mixing in a latent space.  AMR considered in an autoencoder, and Manifold Mixup is also relevant because its theoretical analysis explicitly considers flattening although in a somewhat difference sense (and both are different from what's done here).

  -The object tracking experiments don't seem very convincing to me (just looking at table 2 at least).

Review:

  This paper considers augmenting the hierarchical VHP-VAE with a criteria in which the jacobian is approximately regularized at interpolations and extrapolations between different points in z space.  The experiments suggest this is an important problem with VHP-VAE and also that it's successfully addressed.

**Experience Assessment:**

I have published one or two papers in this area.

**Review Assessment: Checking Correctness Of Derivations And Theory:**

I assessed the sensibility of the derivations and theory.

**Review Assessment: Checking Correctness Of Experiments:**

I assessed the sensibility of the experiments.

**Review Assessment: Thoroughness In Paper Reading:**

I read the paper at least twice and used my best judgement in assessing the paper.

---

> ### Author Response · Authors · 2019-11-13
> **Replies to Reviewer #1**
>
> We would like to thank the reviewer for the thorough and helpful reviews.
>
>
>   -This paper extends VHP-VAE with jacobian regularization, which is approximated (the paper doesn't say so but I think it's a first order taylor expansion).
>
> Answer: Thank you very much for pointing that out. We have added this to the manuscript.
>
>
>   -In figure 2, the background color indicates the degree of magnification (so the VAE-VHP has greater variability in distances?)  I found this figure a bit hard to itnerpret.
>
> Answer: We assume that the reviewer means Fig. 3 instead of Fig. 2. It is more intuitive to interpret the MF by means of the equidistance lines. The shape and size of the equidistance lines are more irregular if the variability distance is larger.
>
>
>   -This paper cites Mixup but there are two more papers to consider here: Manifold Mixup (ICML 2019) and Adversarial Mixup Resynthesis (Neurips 2019) which both considered mixing in a latent space.  AMR considered in an autoencoder, and Manifold Mixup is also relevant because its theoretical analysis explicitly considers flattening although in a somewhat difference sense (and both are different from what's done here).
>
> Answer: Thanks for the suggestion. We have cited these two papers in the related work.
>
>
>   -The object tracking experiments don't seem very convincing to me (just looking at table 2 at least).
>
> Answer: Table 2 shows both supervised learning and unsupervised learning methods. Usually supervised learning methods require labeled data, which is often not possible. Tasks such as autonomous driving are very data hungry and it is expensive to label the required data. Our unsupervised method (no labels required) is close to the supervised learning method (DeepSORT) and outperforms other unsupervised learning models. In four out of 16 metrics, our method outperforms the supervised model; In 11 out of 16 metrics, our model outperforms other non-supervised learning models.  We have revised Table 2 to highlight the results.

---

### Official Review · AnonReviewer2 · 2019-10-18
**Official Blind Review #2**

**Rating:** 1

**Review:**

Summary of paper:
The paper is concerned with the geometry of latent spaces in VAEs. In particular, it is argued that since geodesics (shortest paths) in the Riemannian interpretation of latent spaces are expensive to compute, then it might be beneficial to regularize the decoder (generator) to be flat, such that geodesics are straight lines. One such regularization is proposed.

Review:
I have several concerns with the paper:

1) Geodesics are never motivated:
The paper provides no motivation for why geodesics are interesting objects in the first place, so it is not clear to me what the authors are even trying to approximate.

2) Under the usual motivation, the work is flawed:
The usual motivation for geodesics is that they should follow the trend of the data (e.g. go through regions of high density). Since no other motivation is provided, I will assume this to be the motivation of the paper as well. The paper propose to use a flexible prior and then approximate geodesics by straight lines. Beyond the most simple linear models, then this cannot work. If the prior is flexible, then straight lines will hardly ever constitute paths through regions of high density. The core idea of the work, thus, seem to be in conflict with itself.

3) A substantial bias is ignored:
The paper consider the Riemannian metric associated with the *mean* decoder. Due to regularization, holes in the data manifold will be smoothly interpolated by the mean decoder, such that geodesics under the associated metric will systematically be attracted to holes in the data manifold. Hauberg discuss this issue in great length here:

  https://arxiv.org/abs/1806.04994

Here it is also demonstrated that geodesics under the mean decoder tend to be straight lines (which is also what the authors observe). Taking the stochasticity of the VAE decoder into account drastically change the behavior of geodesics to naturally follow the trend of the data.

4) Related work is mischaracterized:
Previous work on the geometry of latent spaces largely fall into two categories: those that treat the decoder as deterministic and those that treat it as being stochastic. In the cited papers Arvanitidis et al and Tosi et al consider stochastic decoders, while the other consider deterministic decoders. Given that geodesics have significantly different behavior in the two cases, it is odd that the difference is never discussed in the paper.

5) It is not clear to me what the experiments actually show:

-- I did not understand the sentence (page 5): "The model is more invariant if the condition number is smaller..." What does it mean to be "more invariant" ? And how is invariance (to what) related to the condition number of the metric?

-- Figure 3 show example geodesics, but only geodesics going between clusters (I have no idea how such geodesics should look). If I look at the yellow cluster of Fig3a, then it seems clear  to me that geodesics really should be circular arcs, yet this is being approximated with straight lines. Are the ground truth geodesics circular? At the end, it seems like the shown examples are the least informative ones, and that intra-cluster geodesics would carry much more meaning.

-- What am I supposed to learn from the "Smoothness" experiment (page 7) ? My only take-away is currently that the proposed regularization does what it is asked to do. It is not clear to me if what it aims to do is desirable? Does the experiment shed light on the desirability of the regularizer or is it more of a "unit test" that show that the regularizer is correctly implemented?

-- In the "Geodesic" experiment (page 7) I don't agree with the choice of baseline. If I understand correctly, the baseline approximate geodesics with shortest paths over the neighbor graph (akin to Isomap). However, there is no reason to believe that the resulting paths bare any resemblance to geodesics under the studied Riemannian metric. The above-mentioned paper by Hauberg provide significant evidence that these baseline geodesics are not at all related to the actual geodesics of the studied metric. The only sensible baseline I can think of is the expensive optimization-based geodesics.

== rebuttal ==
I have read the rebuttal and discussed with the authors, and I retain my original score.

**Experience Assessment:**

I have published in this field for several years.

**Review Assessment: Checking Correctness Of Derivations And Theory:**

I did not assess the derivations or theory.

**Review Assessment: Checking Correctness Of Experiments:**

I assessed the sensibility of the experiments.

**Review Assessment: Thoroughness In Paper Reading:**

I read the paper at least twice and used my best judgement in assessing the paper.

---

> ### Author Response · Authors · 2019-11-13
> **Title: Replies to Reviewer #2. Part 1**
>
> We thank the reviewer for the valuable comments and suggestions.
>
> 1) Answer: Thanks for pointing this out. We have added a clearer motivation. The main aim of the paper is approximating the geodesic sufficiently (sacrificing a bit of accuracy) but maintain high computational speed to enable useful applications. We use the Riemannian distance (with the geodesic as the shortest path) as an inspiration to develop a distance function which can be computed rapidly (1000 times faster than previous methods, see Sec. 4.2). This is of crucial importance in certain scenarios such as autonomous driving.
>
> 2) Answer: We agree that the geodesic should follow the trend of the data and the variance of the decoder can improve the results. However, our goal is to achieve a homogeneous MF, and consequently Euclidean distance approximates to Riemannian distance. We use data augmentation (mixup) to fill the data into the missing data regions, so that there is no low density regions. While this certainly is a further inductive bias on the data distribution, e.g. it makes low density regions less likely to emerge during training, it is a heuristic that helps the experimental results.
>
> 3) Answer: We agree that moving towards a fully stochastic decoder and using the appropriate geodesics framework is a promising direction. However, we are mainly interested in a distance function. Further, all experiments are done with a Gaussian likelihood with homoscedastic variance. Hence, the stochasticity of the decoder in this work is only used to explain the noise of the data, which certainly is not an interesting thing to reflect for estimating distances.  We have cited the paper mentioned above [Hauberg, 2019] and extend the paper regarding this direction.
>
> As shown in Fig. 3b (green lines), the geodesics do NOT tend to be straight lines. This is only the case because of our contribution (see experiments in Fig. 3a). Therefore, our results are different to [Hauberg, 2019]. Regions in the latent space, which have a high MF are “stretched” by the Jacobian regularisation, and the distance between points is thus established in a different way.
>
> In [Hauberg, 2019], the variance of the Jacobian is used to constrain the geodesic to follow the data manifold. However, there are alternative solutions for deterministic approaches like e.g. regularising the singular term of the SVD decomposition [Chen et al., 2018a].
>
>
> 4) Answer:  We have added the difference in the related work. The main difference is that the stochastic methods work for the regions without data, because the RBF layer generates high MF for those. However, it is less general and post hoc. The uncertainty does not emerge from a principled way (such as in a Bayesian model) but is instead driven by certain assumptions. The deterministic method requires other strategies to guarantee that the geodesic is within the data manifold. In our proposed method, we regularise the latent space to have the same metric, so that we do not need to consider the region without data.

---

> > ### Comment · AnonReviewer2 · 2019-11-14
> > **Quick question**
> >
> > Hi,
> >
> > Thanks for the response; I've had a quick look at the updated manuscript, but I am not able to find the mentioned motivation of why you want to compute geodesics. Can you point to a specific paragraph in case I missed something obvious?
> >
> > To be clear: I am not asking why you want the geodesics to be fast to compute, I'm asking why you want to use geodesic distances in the first place, and which properties you expect a geodesic to posses (it must have some desirable properties since you opt to compute it).
> >
> > Thanks!

---

> > > ### Comment · AnonReviewer1 · 2019-11-14
> > > **Motivation**
> > >
> > > You don't have to like the paper, but I'll try to provide my understanding of the motivation.
> > >
> > > I think that they want to regularize p(x | z) to be "flatter", in the sense that the gradients of x with respect to z are more constant as one moves around z.  They accomplish this with an objective that's related to mixup.
> > >
> > > The paper's argument is then that this regularization allows the choice of a more complex q(z|x), for example a hierarchical model.

---

> > > > ### Comment · AnonReviewer2 · 2019-11-14
> > > > **Right**
> > > >
> > > > That part I understand. I think it seems sensible to regularize towards flatter manifolds (that's what most regularization sets out to achieve). To formalize a notion of flatness the authors go far near-constant volume measures (magnification factors, MF). This aspect of the paper seem sensible to me.
> > > >
> > > > But none of this seem to be related to the notion of geodesic curves, which constitute a significant part of the paper. All I'm trying to understand is where the geodesics fit into all of this. To understand this, I feel I must first understand exactly which properties the authors expect a geodesic to have.

---

> > > > > ### Author Response · Authors · 2019-11-14
> > > > > **Response about geodesics**
> > > > >
> > > > > We expect that short (approximate) geodesics under the learned model indicate similarity of data points in question. We will add this to the last paragraph of the introduction.

---

> > > > > > ### Comment · AnonReviewer2 · 2019-11-15
> > > > > > **Re: geodesics**
> > > > > >
> > > > > > Thanks for being explicit.
> > > > > >
> > > > > > You are saying that you want the geodesic distance to be short for similar data points. Given that you are working with geodesics under the pull-back metric, I take this to mean that you want short geodesic distances between points that are nearby in data space (measured along the manifold), and long for points that are far away from each other in data space.
> > > > > >
> > > > > > When you regularize towards flatness, you (by the very definition of the pull-back) do generally not have this requested property as the geodesic curves must take "shortcuts" through regions where the regularizer is the most active (i.e. where data is lacking). I do not agree with the argument that mixup can be used to avoid this problem (by filling out regions of space with limited data): for example, in motion capture (the example of the authors), period motion (walking, ...) must topologically result in a circular latent space. Filling in regions of space where data is lacking is a topological impossibility.
> > > > > >
> > > > > > While I think the proposed regularization may have value, I have to judge the work by the geodesic aspect as that dominates the paper. Here I see fundamental mistakes (as pointed to in my initial review), and I have not seen a convincing rebuttal from the authors. Hence I retain my score.

---

> > > > > > > ### Author Response · Authors · 2019-11-15
> > > > > > > **Thanks for reviewing**
> > > > > > >
> > > > > > > The reviewer’s arguments are mainly based on the above mentioned paper “only Bayes should learn a manifold” (which is not peer reviewed). We have doubts about some points in that paper. There are alternative solutions (not “only” Bayes), as we mentioned above, to learn a manifold for a deterministic decoder Jacobian.
> > > > > > >
> > > > > > > We have argued that our contribution is a fast, geodesics-inspired distance function based on generative models (we declared in the paper that a bit of geodesic accuracy is sacrificed). It is also demonstrated that it works empirically in relevant settings. We regret that this is not enough reason to refrain from a “clear reject” of the paper in your eyes.

---

> > > > > > > > ### Comment · AnonReviewer2 · 2019-11-15
> > > > > > > > **Misunderstanding**
> > > > > > > >
> > > > > > > > To be clear, my argument is not based on the mentioned "Only Bayes..." paper; I point to this reference as it is the clearest exposition of the problem that I have seen. The fundamental problem is trivial: if you regularize towards smooth manifolds, then distances along the manifold will, by definition, be shorter in regions where the regularization dominates. This bias imply that geodesic distances are not only short when connecting similar points, but may also be short when connecting data points that belong to different components of the manifold.
> > > > > > > >
> > > > > > > > My concern is that you choose to ignore this bias. If you choose to solve the problem using Bayesian methods, SVD regularization or some other means, is from my perspective irrelevant. My issue is that as long as the bias is ignored, then the entire proposed model suffers the consequences.
> > > > > > > >
> > > > > > > > (as a historical side-remark: the original paper from Tosi et al that introduced the idea of Riemannian metrics in latent spaces allude to this bias and point out that the Bayesian solution removes the problem; the argument that "Only Bayes should learn a manifold", thus, actually goes back to the initial paper of the field)

---

> ### Author Response · Authors · 2019-11-13
> **Replies to Reviewer #2. Part 2**
>
> 5)
> 5.1 Question: … What does it mean to be "more invariant" ? And how is invariance (to what) related to the condition number of the metric?
>
> Answer: “More invariant” means the equidistance lines are more similar along different directions and more similar at different centres. As described in section 4 paragraph 2, the conditional number is computed by means of the metric tensor as well as the equidistance lines. We have clarified it in the paper.
>
>
> 5.2 Question: Figure 3 shows example geodesics, but only geodesics going between clusters...
>
> Answer:
> We agree that it does not make sense to interpolate between clusters using the previous models. However, in our task, it is important to have a correct distance in the latent space between clusters, but the interpolation is just a tool to show show the results. Our proposed model corrects the metric between clusters, which makes it possible for interpolation between classes (see Fig. 6 for intuitive results). It is more difficult to interpret interpolation between clusters using image dataset (where there are clear separations between clusters), but more intuitive using MoCap data (where there is a path from each data point to each other data point).
>
> For better observation, the data points (e.g. the yellow dots) are the mean output of the encoder. For the Geodesic without jacobian normalisation (Fig. 3b), it does not necessarily follow the mean of the trajectory. In addition, our method regularises to flatten the latent space, but it cannot completely flatten special data structure such as a circle. However, it already reduces a lot of unsafe (large MF) area. See the circle example in Fig. 1. There is still a tiny high MF area in the center.
>
>
> 5.3 Question: What am I supposed to learn from the "Smoothness" experiment (page 7) ? ...
>
> Answer:
> If it is smooth, a continuous trajectory in the latent space is corresponding to a continuous trajectory in the observation space. In this case (e.g., two clusters would not squeeze together), we can safely measure the distance in the latent space.
>
>
> 5.4 Question: In the "Geodesic" experiment (page 7) I don't agree with the choice of baseline...
>
> Answer:
> As shown in the paper [Chen et al., 2018a; Arvanitidis et al., 2018], the expensive geodesic method has not been developed for latent spaces with more than two dimensions. Therefore, we select the graph-based method for comparison. In addition, the number of graph nodes and neighbours influence the accuracy of the approximated geodesic. Similarly, the accuracy of ODE- and NN-based geodesic approximations depends on the step length as well.

---

### Official Review · AnonReviewer3 · 2019-10-25
**Official Blind Review #3**

**Rating:** 6

**Review:**

1.	The idea of explicitly forcing the encoding space to be flat by putting constraint on metric tensor is simple but neat.
2.	The use of Jacobi regularization in Eq. (9) is effective but the choice of using interpolation to extend this in the entire decoding space is kind of adhoc. Can authors please justify?
3.	Not sure how authors put the Lipschitz continuity constraint on f. Please explain.
4.	The title of “Flat manifold VAEs” is misleading as it potentially means VAEs for flat manifold
5.	I wonder what will happen if you put an unfolding constraint in the encoding space like LLE, ISOMAP etc.. The loss function is data driven so this should give atleast similar behavior.
6.	Overall I like the experimental setup, but the tracking experiment is kind of distracting. The authors may want to remove this experiment.
7.	In Fig. 7, the authors have shown with and without  Jacobi normalization which I am really not convinced with, need better explanation.


**Experience Assessment:**

I have read many papers in this area.

**Review Assessment: Checking Correctness Of Derivations And Theory:**

I assessed the sensibility of the derivations and theory.

**Review Assessment: Checking Correctness Of Experiments:**

I assessed the sensibility of the experiments.

**Review Assessment: Thoroughness In Paper Reading:**

I read the paper at least twice and used my best judgement in assessing the paper.

---

> ### Author Response · Authors · 2019-11-13
> **Replies to Reviewer #3**
>
> We appreciate all reviewers’ opinions and suggestions very much and it enabled us to substantially improve our manuscript.
>
> 2. Answer: If we understand correctly, the question is why we interpolate to extend the data in the entire latent space and input the data into the Jacobian regularisation. When we measure the distance between two points in the latent space, the region (latent space) in between of these points has to be regularised/smooth. To obtain such a regularised latent space, the regulariser needs data at these regions---otherwise the latent space might be folded/unsmooth. However, data is not always available (e.g., data between two clusters). Therefore, we augment the data for the entire latent space using mixup (a powerful and simple method based on interpolation). To constrain the latent space between any two points to be smooth (approximate constant MF), we use interpolated data as input for the Jacobian regularisation term in Eq (9).
>
> 3. Answer: We do not put the Lipschitz continuity constraint on the decoder, but we prove that our decoder satisfies Lipschitz continuity. We have clarified it in the updated version.
>
> 4. Answer: Thanks for pointing this out. We have modified the title to “Learning flat latent manifolds with VAEs”.
>
> 5. Answer: We agree that this would be interesting but consider it beyond the scope of this work.
>
> 6. Answer: This experiment shows how our method is applied to a state-of-the-art algorithm in terms of measuring the distance. We have revised the tracking experiment to improve readability.
>
> 7. Answer: The model can be relatively smooth without Jacobian normalisation, but the distance in the latent space cannot reflect the truth distance. We take Jogging and walking as an example. Jogging is a larger movement than walking in terms of the joints. Without the Jacobian normalisation,  the distribution of walking in the latent space is still larger than that of jogging, which is in conflict with the true distance.

---

### Decision · Program_Chairs · 2019-12-19

**Decision:**

Reject

**Comment:**

The paper proposes to regularize the decoder of the VAE to have a flat pull-back metric, with the goal of making Euclidean distances in the latent space correspond to geodesic distances. This, in turn, results in faster geodesic distance computation. I share the concern of R2 that this regularization towards a flat metric could result in "biased" geodesic distances in regions where data is scarce. I suggest the authors discuss in the next version of the paper if there are situations where this regularization might have drawbacks and if possible, conduct experiments (perhaps on toy data) to either rule out or highlight these points, particularly about scarce data regions.